# Bispecific Antibodies, Nanobodies and Extracellular Vesicles: Present and Future to Cancer Target Therapy

**DOI:** 10.3390/biom15050639

**Published:** 2025-04-29

**Authors:** Asier Lizama-Muñoz, Julio Plaza-Diaz

**Affiliations:** 1Department of Biochemistry, Molecular Biology and Immunology III, Faculty of Medicine, University of Granada, 18016 Granada, Spain; e.asierlizamunoz@go.ugr.es; 2Clinical Analysis and Immunology Department, University Hospital Virgen de las Nieves, 18014 Granada, Spain; 3ANUT-DSM (Alimentaciò, Nutrició Desenvolupament i Salut Mental), Departament de Bioquímica i Biotecnologia, Universitat Rovira i Virgili, 43201 Reus, Spain; 4School of Health Sciences, Universidad Internacional de La Rioja, Avenida de la Paz 137, 26006 Logroño, Spain

**Keywords:** bispecific antibodies, cancer therapy, extracellular vesicles, nanobodies, target therapy

## Abstract

Cancer remains one of the leading causes of mortality worldwide, with a growing need for precise and effective treatments. Traditional therapies such as chemotherapy and radiotherapy have limitations, including off-target effects and drug resistance. In recent years, targeted therapies have emerged as promising alternatives, aiming to improve treatment specificity and reduce systemic toxicity. Among the most innovative approaches, bispecific antibodies, nanobodies, and extracellular vesicles offer distinct and complementary mechanisms for cancer therapy. Bispecific antibodies enhance immune responses and enable dual-targeting of cancer cells, nanobodies provide superior tumor penetration due to their small size, and extracellular vesicles present a novel platform for drug and RNA delivery. This work aims to review and analyze these three approaches, assessing their current applications, advantages, challenges, and future perspectives.

## 1. Introduction

For many years, patients with cancer have had a limited range of available therapies which include surgery, radiotherapy and chemotherapy either individually or in combination [1]. Usually, surgery and radiotherapy have been used on tumors in early stages, without metastasis. Chemotherapy has been employed when the tumor was refractory to surgery and irradiation or metastases were present [2]. The main problem with this type of treatment is that they have a lack of specificity and damage healthy cells, tissues and organs [1]. Also, chemotherapy has other problems like drug resistance, inefficient drug delivery to the tumor site and side effects, among others [1,3].

The last few decades have seen a great deal of research being conducted on the development of new therapies against cancer that can overcome the limitations of traditional approaches [4]. Among the therapies developed are immunotherapy, cancer vaccines, cytokine therapies and oncolytic virus therapies. In spite of this, the most significant progress was made through a special strategy included in immunotherapy, molecular targeted therapy [5,6,7]. It refers to the use of substances or drugs to specifically attack cancer targets to prevent and block tumor progression and growth [8]. The use of this strategy is not without its drawbacks, as it is only effective if the tumor expresses the biomarkers or molecules for which the strategy has been developed, and it also generates drug resistance, side effects and toxicity, but these are less pronounced than chemotherapy [8,9]. Targeted therapy has also resulted in remarkable survival benefits if we compare it with traditional therapies, primarily chemotherapy [9].

Traditionally molecular targeted therapy has been developed using the properties of antibodies, especially the use of monoclonal antibodies (mAbs) [10,11,12]. The evolution and improvement of the mAb are to be the bispecific antibodies (BsAbs), which have gained more interest given their unique and versatile mechanism of action, their ability to bind to different targets with the specificities of two antibodies utilizing only a molecule and for the fact that they are able to mediate therapeutic effects better than mAb [13,14,15]. Their applications are diverse, including cancer therapy. They can be used to directly act on targeted epitopes such as cell surface molecules or soluble factors. They can also play a role in connecting immune cells to tumor cells and thereby be employed for the delivery of therapeutic components [15,16]. Another step in evolution is nanobodies (Nbs), which are the smallest antibodies with activity known, measuring around one-tenth of conventional mAbs, and due to their small size, they can interact better with internal parts of the tumor than mAb or BsAbs [16,17,18]. They also have other advantages over mAbs and BsAbs, including high affinity, stability and solubility [19].

In recent years the focus has also been put on the tumor microenvironment (TME), which involves a complex and highly dynamic signaling network between cancerous cells and the rest of the organism. This set of signals controls some cancer evasion and progression stages [20].

Tumors can secrete a heterogenous population of extracellular vesicles [21] to modify, modulate and change the TME, interfering with immune cells’ activity and anti-tumor therapies, affecting their therapeutic effectiveness and increasing tumor evasion and progression [20,22]. EVs are a kind of nanoparticle secreted by all cell types that carry bioactive elements [5,20,22]. For that reason, they have received considerable attention as a source of minimally invasive biomarkers for cancer diagnosis and prognosis, and as a potential cancer therapy delivery vector [23,24,25].

According to their biogenesis and size, we can distinguish four types: exosomes (50–200 nm); prostasomes (40–500 nm); microvesicles (MVs, 100–1000 nm) and apoptotic bodies (50–4000 nm) [26]. The two main types of EVs investigated and characterized are exosomes and MVs.

Exosomes originate from late endosomes and require endocytosis for their formation and exocytosis for their secretion; MVs do not require exocytosis because they are directly formed from the plasma membrane. They are produced and secreted by cancer cells, and they have a crucial role in tumor evasion and progression [5,20,22].

Recently, studies have demonstrated that B cells can secrete EVs that contain some kinds of antibodies on their surface and deliver their inside contents to the target cells [27,28,29]. Also, some authors have described that it is possible to coat EVs with antibodies. The EVs were modified to achieve the expression of two types of monoclonal antibodies for dual targeting, so instead of two mAbs, it is possible to use a BsAb or Nbs [30]. This potential combination of EVs with antibodies on their surface can better lead EVs to tumor cells and improve selectivity by reducing off-target effects.

The aim of this review is to explore, analyze and evaluate the present and future perspectives, including their current applications, advantages and challenges of BsAbs, Nbs and EVs in cancer target therapy, emphasizing the possibilities of combining them to improve therapy effectiveness.

## 2. Bispecific Antibodies (BsAbs) in Cancer Therapy

BsAbs are a type of antibody with two recognition sites that can join or bind to two distinct antigens, epitopes or molecules [15,31]. The first BsAbs were generated by fusing two types of hybridomas that produced two distinct types of mAbs. However, genetic engineering has made it possible to produce them more easily [15]. The concept of a BsAb was based on a molecule composed of two different heavy chains and two different light chains in the same molecule [16]. Nowadays, there are many BsAbs on molecular platforms that can be divided into two types: IgG-like and non-IgG-like [15]. IgG-like BsAbs are similar to classical IgG molecules but are created by pairing two heterologous polypeptide chains [15,32]. Non-IgG-like BsAbs differ a little from the classical concept of an antibody because they mainly mediate their therapeutic effects through their antigen-binding ability due to their lack of the Fc domain. They are usually created by connecting the variable domains of the heavy and light chains (VH and VL) of an IgG with a peptide linker [32]. They include a large variety of molecular platforms, such as dual-affinity re-targeting antibodies (DARTs), bispecific T or killer-cell engagers (BiTEs or BiKEs), bispecific tandem scFvs or tandem diabodies (TandAbs), Fab-fusion proteins and bi-nanobodies [15,16,32].

Their ability to join two different targets or molecules makes them a great option to be used in cancer target therapy. They have the therapeutic effect of combining two mAbs using only a single molecule. Their capacity is not only limited to blocking surface molecules expressed by tumor cells that are critical to their proliferation or that interfere with immunosurveillance. They can also be used for recruiting, activating and potentiating the immune system recognition and activity against tumor cells [13,14].

### 2.1. BsAb Classification by Their Mechanism of Action

BsAbs can be classified into four main groups, depending on their mechanism of action: the first one blocks two different pathways and usually targets two tumor-associated antigens (TAAs); the second one targets immune checkpoints; the third one aims at the activation and recruitment of the immune system; and the last one blocks inflammatory factors [15]. The last type is used for non-cancer diseases, so it will not be mentioned. As shown in Table 1, the most notable BsAbs are grouped in terms of their mechanism of action and we will discuss them in more detail below.

#### 2.1.1. BsAbs That Block Two Pathways

They usually target two different TAAs; this offers some advantages that include the selectivity of antibody binding, the modulation of two distinct functional pathways at the same time, holding up tumor development, modifying the TME and inhibiting blood vessel creation and regeneration [15,58,59].

There is only one BsAb approved by the FDA that targets two TAA; its name is Amivantamab (JNJ-61186372). It targets the epidermal growth factor receptor (EGFR) and the mesenchymal−epithelial transition factor (MET). It has been used for the treatment of non-small-cell lung cancer (NSCLC), which is refractory to tyrosine kinase inhibitors (TKIs), especially for a subtype of NSCLC with EGFR mutations [13,33,60].

Furthermore, there are also some interesting candidates in clinical trials. Vanucizumab is a BsAb that inhibits vascular endothelial growth factor (VEGF) and angiopoietin-2 (Ang-2). It has demonstrated safety and anti-tumor activity in phase I and II studies for both colorectal and metastatic cancers [15,34]. Navicixizumab is another example of BsAb which in this case, inhibits VEGF and delta-like ligand 4 (DLL4). It successfully completed phase Ib for ovarian cancer [15,35]. Another example of BsAb is Zanidatamab, which binds to two distinct extracellular domains of the human epidermal growth factor receptor 2 (HER2–ECD4 and ECD2) and has completed phase II trials in different solid tumors [15,36].

#### 2.1.2. BsAbs Against Immune Checkpoints

Dual immune checkpoint inhibitors (ICIs) or dual immune-checkpoint blockers (ICBs) look to block two different immune-related molecules such as cytotoxic T lymphocyte-associated molecule 4 (CTLA-4), programmed cell death ligand 1 (PD-L1), programmed cell death receptor 1 (PD-1) and lymphocyte activation gene-3 (LAG-3), among others; because when they are blocked, the consequence is that the T cells can be reactivated [10,11,15,61,62]. A mechanism of tumor evasion is the suppression of the immune system, especially the T cells [63]. For that reason, the ICIs seek the reactivation of T cells to attack the tumor [45,63].

There are a large number of clinical trials focused on the use of BsAbs that can reactivate T cells by blocking the ICIs, primarily for solid tumors but unfortunately, there are not any approved yet [15,37,64]. The most advanced one is Tebotelimab (MGD013) which targets PD-1/LAG-3 and has been demonstrated to be safe for the treatment of multiple cancer types and is now in phase II/III [37]. Lymphocytes that express both PD-1/LAG-3 at the same time are dysfunctional [15,37,38]. Another example is IBI318, a BsAb that targets PD-1/PD-L1 and has completed phase I/II for natural killer (NK) cells/T-cell lymphoma [32,39]. Cadonilimab (AK104) is another example of ICI BsAb which targets PD-1/CTLA-4 and it has completed phase II for adenocarcinoma and phase Ib/II for NSCLC in patients resistant to anti-PD-1/PD-L1 antibodies [15,40,41].

Moreover, BsAbs have also been developed against one immune checkpoint and a TAA. An example of this type of BsAb is Ivonescimab (AK112), which targets PD-1/VEGF and has completed phase I/II for advanced NSCLC [15,42,43].

#### 2.1.3. BsAbs That Aim at the Activation of the Immune System

By activating the immune system, they aim to create bridges between immune cells such as T and NK cells and tumor cells. Its mechanism of action is antibody-dependent cellular cytotoxicity (ADCC) [31]. An example of these types of BsAbs are BiTEs, which usually target T cells with CD3 and one tumoral antigen [61]. They are formed by two scFv fragments of different mAbs joined by a linker [15]. Bispecific killer cell engagers (BiKEs) have a similar structure to BiTEs, but they bind to NK cells instead of T cells [32]. One of the approved BsAbs used in clinics is blinatumomab (MT103), a BiTE that binds to CD3/CD19 and that has been used for the treatment of relapsed or refractory (R/R) Philadelphia chromosome-negative or -positive B cell acute lymphoblastic leukemia (Ph −/+ B-ALL) and non-Hodgkin lymphoma (NHL) [15,32,45]. Based on the success of this BsAb, there have been developed other conceptually based BsAbs for other cancer types. Glofitamab is a BiTE with an innovative 2:1 structure, bivalent to CD20 and monovalent for CD3 [65]; this causes it to activate T cells to kill B malignant cells. Glofitamab is approved for R/R diffuse large B-cell lymphoma (DLBCL) and is in clinical trials for R/R mantle cell lymphoma (MCL) [32,46,47]. Teclistamab is an IgG-like BiTE that recognizes CD3 and the B cell maturation antigen (BCMA). It is used for the treatment of follicular NHL and multiple myeloma (MM) [13,48,60]. An example of a BiKEs is Acimtamig (AFM13), which is a TandAb that can recognize simultaneously CD30/CD16a with two binding sites for each antigen without an Fc domain. It is in phase II of clinical trials for R/R peripheral T-cell lymphoma (PTCL) [53,54,55]. Another example is E5C1 which binds to CD16a/HER2 and has demonstrated efficacy in preclinical trials on cell culture and on a mouse model of ovarian metastatic cancer [56,57].

BiTEs or BiKEs are only a class of BsAbs included in this group. There are some other types of BsAbs that aim at the activation of the immune system such as trifunctional bispecific antibodies (trAbs or Triomabs) or immune-mobilizing monoclonal T-cell receptors against cancer (ImmTACs) [66]. An example of a trAbs is catumaxomab, an IgG-like BsAb that binds CD3 and the epithelial cell adhesion molecule (EpCAM) simultaneously [67]. As it contains a functional Fc region, it can bind to an Fc gamma receptor (FcγR) present in some innate cells like monocytes/macrophages, dendritic cells (DC) or NK cells. It is approved for the intraperitoneal treatment of malignant ascites [15,49,50]. And an example of an ImmTAC is Tebentafusp. This is composed of an anti-CD3 single-chain variable fragment joined to a soluble affinity-enhanced HLA-A*02:01 restricted TCR-like molecule. It is approved for the treatment of uveal melanoma [51,52].

### 2.2. Advantages and Limitations

The main advantages of BsAb are their higher specificity compared with mAbs: they can mediate therapeutic effects better than mAbs, their capacity to recognize two different antigens with the specificity of a mAb but using only one molecule and their ability to recognize tumor cells more sensitively and activate the immune system at the same time [13,14,15,31].

Nevertheless, they have some limitations including their immunogenicity because some of them are produced in animals like mice or hamsters, impurities and chimeric origin; off-target effects on the immune system caused by their nonspecific activation, highlighting the need to develop more specific activation mechanisms; and the cost of production because they usually need a very complex and costly production process to produce a purified molecule [68]. Their poor penetration and infiltration into the tumor and immunosuppressive TME are other common problems that can be solved using non-IgG-like or smaller antibodies [31,60,69,70]. In Table 2, we summarize the main advantages and disadvantages of BsAbs.

## 3. Nanobodies (Nbs) in Cancer Therapy

Nanobodies (Nbs) are also called single-domain antibodies (sdAbs) or variable domain of heavy-chain antibodies (VHHs) because they are composed of an scFv derived from the heavy-chain antibodies of the *Camelidae* family (usually, camels and llamas) and some fish (in particular, sharks) [16,19,71,72]. They are the smallest natural antigen-binding fragment known. Their dimensions are on the nanometer scale, with approximately 2.5 nm in diameter and 4 nm in height, and a molecular weight of about ten parts of a classical mAb, approximately 15 kD [16,21,71,73]. They are formed by three hypervariable antigen-binding loops (H1, H2 and H3) or complementarity-determining regions (CDR1/2/3) and four conserved framework regions (FR1/2/3/4). A disulfide bond between CDR1 and CDR3 increases their stability [16,21,72,73,74].

They have substantial benefits, compared with mAbs and BsAbs, that include their small size, high affinity, stability, solubility and the ability to target epitopes inaccessible to other types of antibodies [19,71,74]. An important feature of their benefits is that they have low immunogenicity because they share a sequence with the human type 3 VH regions [19,71]. Due to their small size, they can rapidly extravasate from the blood, be quickly eliminated by the renal excretion system and also be able to cross the blood–brain barrier (BBB) [71,73]. Moreover, they have unusual physical properties that include prolonged stability at storage conditions (+4 °C and −20 °C), tolerance to increased temperature and resistance to proteolytic degradation, non-physiological pH and chemical denaturants [21,75]. There are three main strategies to produce Nbs that are based on the creation of immune, naïve or synthetic libraries. The first relies on immunizing a camelid with the molecule or antigen of interest and the collection of a small amount of blood to purify the Nbs; the second on collecting higher amounts of blood from several healthy camelids that were not intentionally immunized; and, the last, on creating Nbs by randomly changing CDR regions of a previous available structure and selecting the Nbs based on their favorable properties [72]. The benefit of the third strategy is that it can be carried out with genetic engineering using molecular biology and recombinant DNA technology without the need for an animal [72,73]. Then, they can also be easily produced in microorganisms (such as bacteria or yeast) or in eukaryotic systems (like mammalian or plant cells) reducing their production cost and minimizing production-associated costs [19,21,71].

As antibodies, they can directly bind to a receptor or surface molecule to block it or produce an effect on the target cell, but they can also be functionalized by joining some other molecules or drugs to specifically mark or attack tumor cells [16]. Their unique structure and properties enable them to be used in different research, therapeutic and diagnostic applications that include biosensing, targeting surface receptors, imaging, affinity-capturing and theranostics [71,74].

### 3.1. Current Applications in Cancer Therapy

For cancer therapy, we distinguish two main applications of Nbs: for imaging and diagnosis, and for cancer treatment as a therapeutic agent. Nbs can be designed for targeting different surface molecules that include TAAs like HER2, EGFR, VEGF; T cell, B cell, NK or macrophage receptors; immune checkpoints such as PD-1, PD-L1 and CTLA-4; or any other receptors expressed by targeted cells as well as cytokines or chemokines [21,71,72,73,74]. Table 3 includes some examples of Nbs used for imaging and diagnosis, as well as for cancer treatment.

#### 3.1.1. Nbs for Imaging and Diagnosis

Molecular imaging is a crucial tool for cancer diagnosis and therapy [88]. Diagnosing cancer in its early stages needs an agent for molecular imaging that can penetrate the tumor and bind specifically to it as well as be able to eliminate it rapidly from the body. Nbs are the ideal molecules for this application [21]. They are being developed for use in positron emission tomography (PET), single photon emission computed tomography (SPECT), near-infrared fluorescence imaging (NIR) and ultrasound-based molecular imaging [21,88]. PET imaging is based on positron-emitting radiotracers which can be detected with a PET scanner. Nbs are able to be conjugated with a radioisotope like ^18^F, ^64^Cu, ^68^Ga, ^89^Zr or ^131^I to create a radiotracer. Some examples of Nbs created for PET imaging are ^131^I-SGMIB anti-HER2, ^18^F-AlF-RESCA-MIRC213 anti-HER2 and ^68^Ga-NODAGA-NM-01 anti-PD-L1 [21,76,77,78,88]. SPECT imaging uses gamma-emitting radioisotopes and Nbs are usually labeled to ^99m^Tc. Some examples are ^99m^Tc-PHG102 anti-CLDN18.2 (claudin 18.2), ^99m^Tc-MY6349 anti-Trop2 (trophoblast cell-surface antigen 2) and ^99m^Tc-K2 anti-PD-L1 [79,80,81,88]. NIR imaging is a fluorescence-based method of diagnosis based on molecular light absorption and emission. It has the advantage that it does not use radioisotopes for diagnosis; therefore, it is safer. Nbs are frequently conjugated to IRDye 800CW or IRDye 680RD. An example of this technology-based Nb is E8-IR800CW which targets Cadherin 17 (CDH17) [82,88].

Unfortunately, there are not any Nb approved yet for molecular imaging, but there are some promising candidates that are now in different phases of clinical trials [88].

#### 3.1.2. Nbs for Cancer Treatment

As Nbs combine the properties of a small molecule and the antibodies’ specificity, they are proposed to be a great agent for the development of novel therapeutic strategies [21]. They can be used as ICI to attach classical immune checkpoint molecules such as PD-1, PD-L1 or CTLA4, but they are also able to be used for targeting other surface receptors or molecules that have the ability to promote immune system activity [89]. For example, they can be used to inhibit CD47, an antiphagocytic ligand exposed by tumor cells to evade immune system recognition; to block macrophage receptors for “do not eat me” ligands including the signal regulatory protein-a (SIRP-a), leukocyte immunoglobulin-like receptor B 1/2 (LILRB1/2) and Sialic acid-binding Ig-like lectin 10 (SIGLEC10); to activate the T cell by joining to CD3, CD28 or 4-1BB and promoting T cell proliferation and function; and, to block soluble factors such as CCL2, CLL5, CXCL12, CSF-1 and VEGF-A given off by tumor cells to create an immunosuppressive TME [90]. Nbs have also the ability to be used to introduce them as chimeric antigen receptor constructs in T cells or NK creating CAR-T or CAR-NK and blocking some surface receptors expressed by tumor cells that are crucial to tumor signaling and survival like EGFR, HER2, VEGF receptor (VEGFR), c-Met and CXCR7 [19,21,71,72,73,74,88]. Furthermore, there are other applications for Nbs in nanomedicine. For example, they can also be used for targeted radionuclide therapy (TRNT) [91], which aims to deliver radioisotopes joined to the Nb to kill local cancer cells; as antibody-drug conjugates (ADCs), this strategy combines the targeting efficiency of Nbs with the action of the cytotoxic molecule conjugated to it; as nanobody-albumin nanoparticles (NANAPs), which are composed by an albumin core modified that carries on its surface Nbs conjugated to polyethylene glycol (PEG); or as surface receptors for some kinds of lipid vesicles (liposomes, micelles, etc.) to enhance their binding specificity when they are used as drug delivery systems [72,88].

Moreover, advances in molecular engineering have allowed the development of multifunctional constructs, such as bispecific or trispecific Nbs which can simultaneously bind to multiple immune checkpoints or in combination with other therapies [74]. For example, they act as BiTEs or BiKEs, by joining a tumor cell and an immune cell [71].

### 3.2. Limitations and Future Prospects

They have some disadvantages that are derived from their small size which include the short half-life time in the human body because they are generally rapidly eliminated from the blood by the kidneys. For that reason, another problem with them is their accumulation in the kidneys which limits their potential use [21,74]. Another important issue is potential immunogenicity, especially when they come from camelids but due to their small size, they are less immunogenic than mAbs or BsAbs. Also, the optimal dose that is administered is an important challenge in their application to the clinic. As mAbs or BsAbs, Nbs has some off-target effects and may generate resistance [74]. Some of these problems could be solved by using synthetic or semisynthetic libraries or humanizing Nbs [72]. But combining them with artificial intelligence (AI) may help to solve all the problems encountered, reducing Nbs’ immunogenicity, increasing efficacy and creating easier synthesis strategies [16,73]. Compared with other monoclonal antibodies, such as classical mAbs or BsAbs, Nbs are better tolerated and have much potential for their application in cancer target therapy. Overall, Nbs are a promising strategy for theranostic applications due to their unique physico-chemical properties, small size and easy modification [21,71,72,73,74,88]. Table 4 summarizes the main advantages and disadvantages of Nbs.

## 4. Extracellular Vesicles as a Drug Delivery Platform

EVs is a commonly used term to identify a family of nanoparticles that are membrane-bound small bilayers secreted by all cells that carry some bioactive elements such as proteins, lipids, metabolites and nucleic acids, including immunomodulatory molecules such as cytokines, chemokines and cytotoxic proteins [5,20,22]. Tumors can secrete a heterogenous population of EVs to modify, modulate and change the TME, interfering with immune cells’ activity and anti-cancer therapies, affecting therapeutic effectiveness and increasing tumor evasion and progression [20,22]. We can differentiate four main species of EVs according to their size and biogenesis mechanism: the smallest, which are exosomes (50–200 nm); prostasomes (40–500 nm); microvesicles (MVs, 100–1000 nm); and the biggest, which are apoptotic bodies (50–4000 nm) [26]. The two main types of EVs produced and secreted by cancer cells that are involved in tumor evasion and progression are exosomes and MVs. For that reason, we focused on them [5,20,22,26].

Exosomes originate from the late endosome and their biosynthesis begins with endocytosis, followed by early endosome creation and multivesicular body (MVB) formation [5,20,92,93]. If MVBs merge with lysosomes, it results in their degradation. The remaining endosomes can migrate to form late endosomes which accumulate different intraluminal vesicles (ILVs) inside them. During this process, cytosolic proteins, nucleic acids and lipids are sorted into them [5,92,93]. In their biogenesis the Ras-associated binding (RAB) GTPases proteins and the endosomal sorting complex required for transport (ESCRT) are also involved in controlling this process. ESCRT guides ILVs to the plasma membrane where the endosomal membrane fuses with it to produce exocytosis of exosomes [5,20,22,93,94]. An alternative biogenesis process has been proposed, which is the ESCRT-independent and tetraspanin-dependent process which also involves the syndecan–syntenin–ALIX complex [94,95]. It has been described that exosomes are able to encapsulate a wide variety of molecules that include different classes of RNAs (tRNA, mRNA, miRNAs, pre-miRNAs and other non-coding RNAs), lipids, cytokines, chemokines, cytotoxic proteins and other bioactive molecules that can have an effect on the activity of other cells, especially immune cells [5,20,95]. Exosomes usually contain diverse types of specific surface markers such as tetraspanins (CD9, CD63 and CD81), heat shock proteins (Hsp60, Hsp70 and Hsp90), MVB synthesis-related proteins (ESCRT-III and syndecan–syntenin–ALIX) and membrane transporters and fusion proteins [5,26,95]. These surface molecules are essential for initial binding to the target cell by facilitating initial adhesion [5,22].

MV biogenesis is a less characterized mechanism than exosomes [5]. MV formation does not require exocytosis because they are directly produced by the shedding or budding of the plasma membrane under normal conditions or in response to external stimuli or stress factors [22,26,94,95]. For that reason, the composition of the membrane closely resembles the plasma membrane of the production cell [22]. They commonly contain different types of proteins, nucleic acids, lipids, metabolites and cytoplasmic material [95]. Several proteins have been identified to be MV specific such as CD40/CD40L, ADP-ribosylation factor 6 (ARF6), selectins, phosphatidylserine (PS), Flotillin-2 and Rho family members [26,95]. Moreover, MVs can also internalize via surface receptors or ligands including integrins, tetraspanins and heat-shock proteins [22].

EVs are involved in cell–cell communication, not only between cancer cells themselves or with other cells, but they also play an important role in healthy cell–cell communication; furthermore, they participate in intracellular signal transduction regulation through surface molecules and molecular cargo [5,95]. Pro-inflammatory EVs usually contain cytokines and other DAPMs that induce macrophage polarization, T cell differentiation and leukocyte chemotaxis. In contrast to that, anti-inflammatory EVs have a role in reducing complement and other pro-inflammatory factors [92].

In cancer, EVs also have an important function and are related to cancer cell-induced angiogenesis, and promotion of cancer cell growth and metastasis by upregulating and delivering angiogenic proteins, growth factors and matrix remodeling proteins [96]. One of the most important roles of EVs in cancer is TME modulation, essentially by suppressing immune system activity against tumors and creating a more favorable environment for tumor cells’ development [5,26,92,93,94,95]. They produce a diverse population of exosomes that have a direct effect on all immune system cells through a wide variety of mechanisms that include polarization of innate cells to tumor-associated macrophages (TAMs) and tumor-associated neutrophils (TANs), inhibiting some cells such as DC or lymphocytes T, upregulating Treg, suppressing NK cells by downregulation of some receptors like CD107a or NKG2D, and inducing myeloid-derived suppressor cells (MDSCs) [5,26,93,95]. In addition, EVs miRNAs, lncRNAs and oncoproteins attenuate the expression of some target genes by activating the mitogen-activated protein kinase (MAPK) pathway and the phosphatidylinositol 3-kinase (PI3K)/AKT/mammalian target of rapamycin (mTOR)-PI3K/AKT/mTOR-pathway [93]. Several studies revealed that exosomes are involved in the modulation of chemosensitivity by transferring the resistant phenotype to recipient cells [95]. Exosomes that contain PS are also related to tumor resistance to immunotherapy [5]. A special type of MVs, oncosomes which are produced by primary cancer cells, contain carcinogenic molecules that cause morphological deformation and increase malignant growth in recipient cancer cells [95].

### 4.1. Applications in Targeted Cancer Therapy

EVs are ideal candidates for drug development and delivery due to their biological origin as well as for nucleic acid delivery, as a natural carrier of RNA and proteins [92,93,97]. They have great biocompatibility, low immunogenicity, high stability, targeting capabilities and scalability [92,97]. Another benefit that EVs have is that they are able to cross biological barriers such as the BBB, the mucosal barrier and the placenta [20]. Their high stability and long half-life are mediated by the expression of some “do not eat me” molecules on their surface and this is an advantage to deliver drugs because they offer more biodisponibility than drug delivery alone [97]. It is important to consider that although all cells can produce EVs, not all are suitable for effective drug carriage [92].

We can differentiate two major processes for loading cargos, either small molecules or drugs and nucleic acids (miRNAs or non-coding RNAs) into EVs: pre-loading methods, in which the molecules are introduced without disrupting the vesicle membrane integrity, and post-loading methods, where cargos are added after EV isolation [98]. Pre-loading methods use two strategies. The first one is based on incubating the cells with the substances to be introduced and the cells will produce EVs loaded with them. This technique has been used for the encapsulation of some drugs such as paclitaxel and doxorubicin [98,99]. The second strategy is carried out by doing genetic modifications to production cells to achieve overexpression of the molecule of interest in them [98]. In post-loading methods, the first critical step to introducing cargo into EVs is bypassing the membrane [100]. Therapeutic agents can be incorporated into EVs by using two basic strategies: active and passive encapsulation [99]. Passive methods are quite simple. Encapsulation can easily be achieved by incubating the EVs with hydrophilic molecules or drugs. They enter the EVs by passive diffusion due to the concentration gradient created between the inside and outside of the vesicles [98,99]. Active loading methods are based on the use of extrinsic forces to introduce the drug into the EV. Here, there is a wide variety of processes that include techniques such as sonication, extrusion, freeze and thaw cycles, incubation with membrane permeabilizers, direct conjugation and the use of Abs on some surface molecules [99]. One of the most common methods of introducing small molecules, especially nucleic acids, is electroporation. This technique creates temporary small pores in the membrane through the application of an electrical field that disturbs the phospholipid bilayer [22,93,98,99,100]. EVs can also be created by a synthetic process and for that reason, they are able to be loaded during their synthesis process [100].

The source of EVs is also an important factor to consider. The most used cell sources include cancer cells, immune system cells, mesenchymal stem cells (MSCs) and some commercial cell lines. The right selection of the appropriate cell type is so important because EVs inherit most characteristics of parent cells, such as membrane characteristics and surface molecules, which influence their functions and effects on target cells [101]. EVs form immune cells retaining most of their parent cells’ characteristics such as immune, immunostimulatory and immunoregulatory properties that allow them to inhibit tumor proliferation, improve immune system activity and chemotherapeutic effects, induce cells to be highly immunogenic by increasing recognition of ligands, and stimulate T cell antigen-specific responses. They are also able to be used as a delivery system for exogenous cargo to the target site [101,102,103,104]. Plant EVs are rich in bioactive components and have high biocompatibility, stability and biodegradability with low immunogenicity and toxicity and natural anti-inflammatory and immunomodulatory properties. For that reason, they are very useful for improving immunity function, modulating gene expression and for inducing tumor cell death by apoptosis as well as a delivery system for both hydrophilic and hydrophobic therapeutic agents [101,102,105,106]. EVs derived from animals are usually formed from milk and stand out for their abundance, easy extraction, safety and simple functionalization. They can be used for enhancing immune system activity as a delivery system and also for inhibiting tumor cell progression because they have inherent anti-cancer potential [101,102,107,108]. Tumor cell EVs (TEVs) have been focalized in the current research because they can be used for homotypic targeting, due to their same origin as tumor cells. This provides them with the ability to be self-recognized and absorbed easily, facilitating drug delivery to cancer cells and opening opportunities for personalized medicine [102,109]. This strategy improved significantly, targeting tumor cells, and also infiltration and cell apoptosis induction, reducing toxicity and slide effects on cancer treatment and reaching places that are hard to access such as the central nervous system. In recent years, there has been huge evidence that using them as drug carriers can improve the efficacy of cancer treatment [101,102,110].

The first approach to cancer treatment therapies was based on EVs derived from DC due to their ability to stimulate specific immune responses to tumor cells [92,97]. Other approaches are founded on NK-derived EVs because they contain some pro-apoptotic ligands, macrophages and monocytes-derived EVs due to their ability to transfer cytokines and chemokines [92,97], bovine milk-derived EVs or exosomes, and also EVs from plants [20,92]. Other strategies developed are based on the use of tumor-derived EVs modified by eliminating some immunosuppressive molecules or irradiating them to enhance phagocytosis and T cell infiltration and activity [97]. EVs loaded with some proinflammatory molecules have also been used to potentiate immune responses [92]. Another strategy is based on the depletion or elimination of circulating EVs produced by cancer cells to reduce their immunosuppressive effect and potentiate the immune system’s recognition of the tumor [26,92,97]. Furthermore, some researchers tried to block the EV production by tumor cells to reduce the immunosuppressive TME [26]. Exosomes coating anti-cancer drugs have demonstrated their potential to reduce side effects and enhance treatment efficiency [20]. There are some examples of EV-derived candidates in preclinical or even in clinical trials right now [111,112,113]. Some of them are recapitulated in Table 5. An example of this success is the work carried out by Zhou et al., where they used fresh milk-derived exosomes to deliver the anticancer drug, cisplatin, in a resistant ovarian carcinoma to cisplatin chemotherapeutic treatment [114]. There are some other examples that include the use of exosomes from a diverse cell type loaded with different drugs or molecules [115,116,117], and also, there are some trials that use plant-derived exosomes too (NCT01294072 and NCT01668849, from https://clinicaltrials.gov); not all are exosomes. There are also MVs loaded with chemotherapeutic agents like methotrexate (MTX) as an anti-cancer therapy [118].

### 4.2. Challenges and Future Directions

Despite the fact that EVs are a potential vehicle to deliver drugs or other molecules to modulate the receiving cell and their low immunogenicity due to their biological origin, they have some challenges that should be solved to improve their application to cancer target therapy.

The main problem is the heterogeneity that exists between EVs because they are different depending on the source of them. Their composition is influenced by the production cell, the state of the cell and environmental conditions [20,98,99]. Another challenge is to achieve large-scale production of EVs because, for their application to the clinic, we need good manufacturing practice protocols and a standardization method of production, isolation and also cargo loading to obtain high-purity EVs without disrupting their integrity and properties [20,22,92,93,94,98,99]. Currently available techniques for isolation such as ultracentrifugation and size-exclusion chromatography are yet inconsistent making them poorly reproducible [20,99]. Moreover, the currently available methods for cargo are quite variable, for example, the pre-loading methods’ productivity is rather low, and most of the post-loading methods could harm the stability and integrity of exosomes [98]. One more pending challenge is to clarify if surface modifications can enhance the EVs targeting abilities and therapeutic outcomes or if they affect their integrity, structure and function [22,98].

However, today EVs are better investigated and positioned for diagnostic and prognostic than treatment but they are also a promising vehicle for molecular delivery of cancer target therapy [97]. Table 6 summarizes the main advantages and disadvantages of EVs.

The main information expressed in the manuscript is summarized in Figure 1.

## 5. Comparative Discussion

### 5.1. Comparing BsAbs, Nbs and EVs in Cancer Therapy

While BsAbs and Nbs are based on antibodies binding specificity to interact with a molecule that allows the immune system to react to the tumor, EVs fundamentally act as a natural delivery system for different types of particles that can modulate TME and enhance tumor recognition by the immune system [119]. The best advantage of BsAbs over Nbs is that they are able to join two independent antigens or molecules using only one structure, but Nbs are able to penetrate more easily into the tumor due to their small size [15]. For that reason, some BsAbs are created by linking or putting together two Nbs separated by a peptide linker. As Nbs, EVs due to their small size and biological origin, are able to access the TME while BsAbs sometimes have problems reaching the tumor [120]. Undoubtedly, despite their endogenous origin, EVs are the least immunogenic strategy to access the tumor, but they need to be loaded or functionalized with other molecules to have activity [121]. In contrast, Nbs and BsAbs are intrinsically active due to their structure. EVs are the cheapest ones related to production costs because they are naturally produced by all cell types, it is true that not all cells are suitable for their production, but they are easily produced; Nbs have also a low production cost, but they need an expensive development and engineering process before their synthesis, but when it is obtained, they can be easily produced in different cells [122]; BsAbs also needs a development and engineering process to achieve their structure and function but they cannot be produced as simple as Nbs or EVs. Nevertheless, the main problem with all of them is the off-target effects that they produce, even if they are meticulously designed with high tumor specificity. Table 7 summarizes the main characteristics of BsAbs, Nbs and EVs.

### 5.2. Potential Combinations

Given their potential alone in cancer target therapy, if two of them are combined, such as BsAbs with EVs and Nbs with EVs, the result will reduce their problems. In addition, it will potentiate their activity against the tumor. These combinations are not new. Some authors have before tried to combine the potential of EVs with BsAbs or Nbs [123].

An example of successful binding between BsAbs and EVs is the synthetic multivalent antibodies retargeting exosomes (SMART-Exos) developed by Shi et al. They designed an EV that expresses a BsAb on its surface through genetic modification of its production cells. They developed SMART-Exos technology for breast cancer as an innovative class of immunotherapy for cancer treatment [22,124]. They built two different combinations of EVs with BsAbs. The first one developed was composed of a BsAb constructed by joining an anti-CD3 scFv from UCHT1 antibody to an anti-EGFR scFv from cetuximab with a flexible linker and this structure was fused to an exosomal membrane protein [125,126]. The second one was composed of an anti-HER2 scFv from trastuzumab joined to an anti-CD3 scFv from UCHT1 antibody, as same as the previous one [127]. These two examples represent that it is possible to achieve the expression of a BsAb at the surface of an EV. There are other possibilities to obtain EVs loaded with BsAbs at the membrane with covalent or non-covalent modifications or by using some structure naturally present in EVs [22,128]. This strategy represents an innovative way to merge the properties of EVs as a cargo structure for carrying molecules with the recognition abilities of BsAbs. With this approach, it is possible to enhance the efficiency and efficacy of drug delivery using EVs as a vehicle at the same time that it is also achievable to improve tumor recognition by the immune system due to the dual binding provided by the BsAb.

For the merger of Nbs with EVs, we have several examples of successful ideas. For example, Pham et al. prove that EVs coated with an Nb can enhance the specific delivery of therapeutic molecules to cancer cells that express the appropriate ligands, as a result of increasing drug target efficacy as well as decreasing side effects [129]. More recently, Chen et al., in 2024, developed a derived EV functionalized with an Nb and they demonstrated that this construct was better than the EV alone and significantly enhanced the specific accumulation in tumor cells [130]. The same year, Zhang et al. created a bovine milk exosome functionalized with an EGFR-specific Nb for delivering doxorubicin to cancer cells that express EGFR. They proved that this assembly enhanced EGFR-dependent cell killing while it reduced non-specific toxicity in EGFR-negative cells [131]. These examples represent how EVs coated or functionalized with Nbs were more effective and selective to tumors than other constructs. They combine the properties of Nbs and EVs to enhance their activity and they can access the TME more easily than other structures. They are also an appropriate system to deliver drugs to the tumor site specifically reducing side and off-target effects.

### 5.3. Regulatory and Translational Challenges

It is true that regulatory agencies (FDA or EMA) should watch over the security and efficacy of new therapies developed that wait for approval. For that reason, there is a need to make huge and exhaustive preclinical phases to prove that the biomolecule created is useful and binds to the specific target with low off-target effects. A potential drug must undergo extensive clinical trials to ensure its safety and efficacy, even if it is effective in preclinical stages. Firstly, it is needed to demonstrate that the product is safe and then, whether it is better than the current available options or if it can be used for other portions of patients, for example, for non-responders. All these procedures make it difficult to achieve their real clinical application even if they have promising results in the laboratory [132]. For translational development, all of them (BsAbs, Nbs, EVs and the potential combinations between them) have the same important problems in achieving the approval of regulatory agencies (FDA or EMA). The key issues are related to their limited TME penetration capacity, short plasma half-life, biodisponibility and biodistribution, safety (including immunogenicity) and stability, large-scale production and manufacturing, off-target effects, drug resistance and cost-effectiveness in comparison to available therapies [133,134].

### 5.4. Clinical Considerations and Translational Barriers

BsAbs are frequently used in clinical diagnostics and treatment. For diagnosis, BsAbs can be combined with horseradish peroxidase, used in pre-targeting strategies to aid in clinical diagnosis, and provide better imaging to aid in the early detection, diagnosis and treatment of cancer. Therapeutic strategies primarily aim to precisely target and reactivate immune cells, regulate their activation, fine-tune their fate and function, improve their tolerance, and promote a return to immune homeostasis. Besides treating tumors, BsAbs are also effective in treating hemophilia A, diabetes, Alzheimer’s disease and ophthalmological conditions [15]. There are several challenges associated with translational research of BsAbs: complex design and production, high immunogenicity risk, difficult analytical characterization and an evolving regulatory environment.

Nbs are characterized by good tumor permeability and rapid renal clearance, which enables rapid and sensitive imaging of target tissue within hours of injection. As a result, they meet the criteria for an effective in vivo tracer [135,136,137]. In addition to their enhanced sensitivity, Nbs conjugates can deliver radionuclides to tumors, therefore allowing high-resolution as well as quantitative imaging of tumors. A further advantage of Nbs for nuclear imaging applications is their short biological half-life, comparable to PET isotopes [138,139]. Due to the multitude of advantages that Nbs possess over conventional mAbs and conventional mAb fragments, many alternate methods and techniques are being developed for producing application/situation-dependent Nbs. As a result, Nbs have found great success in various diagnostic formats, such as lateral flow immunoassay, diagnostic ELISAs, biosensors and in vivo diagnostic imaging, not only for the detection of disease but also for the detection of food-borne pathogens and environmental toxins. Additionally, the multiple therapeutic approvals for Nb treatment of cancer and autoimmune diseases have sparked commercial and industrial interest in Nbs over the past few years [17]. In order to conduct translational research on Nbs, several challenges must be overcome. This includes a small size that is excellent for tissue penetration and rapid clearance might require an extension of half-life and lower immunogenicity (camelid origin).

The clinical translation of EVs is still in progress, so there are several areas in which improvements could be made to ensure successful and timely translation into the clinic. Due to the limitations of current techniques for tracking and imaging EVs within cells and tissues, it would be beneficial to develop EV tracers that would improve our understanding of extracellular and intracellular trafficking, which is crucial for gaining mechanistic insights [140,141]. EV subpopulations may be characterized more accurately and rapidly using recently developed single particle analysis techniques, such as Exoview [142] and total internal reflection fluorescence microscopy [143]. Since the heterogeneity of EVs is an essential challenge for regulatory approval, nanoscale methods like nanoflow cytometry [144], imaging flow cytometry [145] and affinity-based methods [146] can be used to characterize and separate subpopulations of EVs to improve therapeutic and diagnostic capabilities and reduce heterogeneity. Translational research on EVs faces several challenges: Insufficient standardized isolation methods, difficult scaling-up, ambiguous potency of assays and regulatory classification challenges.

## 6. Conclusions and Future Perspectives

As a result of the successful development of a variety of BsAbs, successful templates and development experiences have been developed. It is essential to maintain the efficacy of combination therapy and reduce the associated toxicity of combination therapy by ensuring proper structural design and target selection. This is one of the key advantages of BsAbs. Developing novel therapeutic antibodies can be facilitated by systematically understanding excellent examples of BsAbs design [119]. The synergistic combination of advanced library construction, display technologies and AI-guided design will enable Nbs to become powerful research, diagnostic and therapeutic tools.

A transformative step in the future of cancer therapy will be the integration of BsAbs, Nbs and EVs into personalized medicine. We are beginning to understand that one-size-fits-all approaches are insufficient for optimal clinical outcomes as we gain a deeper understanding of tumor heterogeneity. The ability of BsAbs to engage both tumor-specific antigens and immune effector cells offers a promising platform for designing highly tailored treatments based on the molecular profile of a patient’s tumor. The advantages of Nbs include enhanced tissue penetration, reduced immunogenicity and ease of genetic engineering, making them ideal candidates for patient-specific targeting.

Furthermore, EVs are gaining attention as natural, biocompatible carriers for the delivery of antibody fragments, RNA molecules or therapeutic payloads directly to the TME. Due to their inherent ability to reflect the biological state of their parent cells, they are potential tools not only for therapeutic use, but also for diagnostic and monitoring purposes. With the convergence of these technologies and high-throughput omics data, as well as AI-driven patient stratification tools, we may be able to develop customized therapeutic regimens that maximize efficacy while minimizing adverse effects.

These biological therapies are poised to play an increasingly important role in the realization of truly personalized cancer care, providing patients with precision, adaptability and improved outcomes.

## Figures and Tables

**Figure 1 biomolecules-15-00639-f001:**
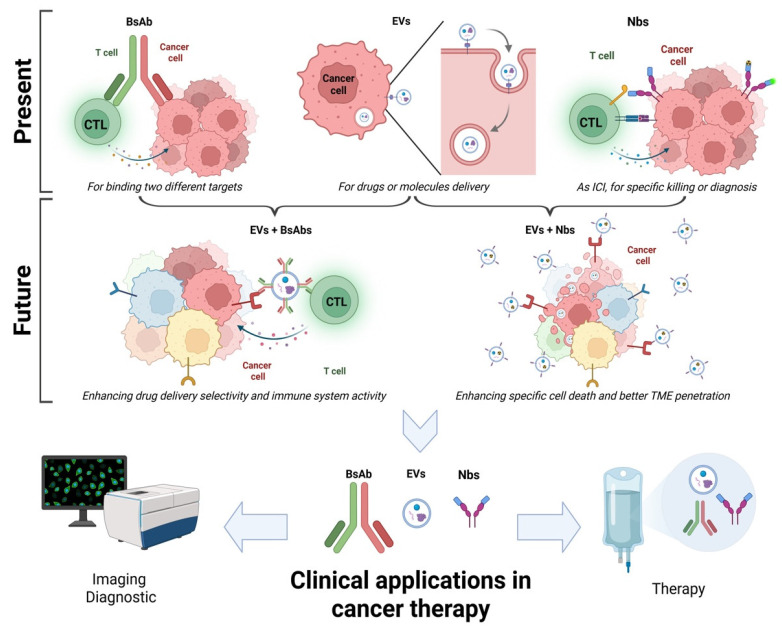
The present and future use of BsAbs, Nbs and EVs in cancer therapy.

**Table 1 biomolecules-15-00639-t001:** Summary of BsAbs that are approved or better positioned in clinical trials.

BsAbs Mechanism of Action	Name	Target 1	Target 2	Application	Phase	Reference
Blocking two pathways	Amivantamab	EGFR	MET	NSCLC with EGFR mutations	Approved	[33]
	Vanucizumab	VEGF	Ang-2	Colorectal and metastatic cancers	Phase I/II	[34]
	Navicixizumab		DLL4	Ovarian cancer	Phase Ib	[35]
	Zanidatamab	HER2	HER2	Solid tumors	Phase II	[36]
Against immune checkpoints	Tebotelimab	PD-1	LAG-3	Multiple cancer types	Phase II/III	[37,38]
	IBI318		PD-L1	NK/T-cell lymphoma	Phase I/II	[39]
	Cadonilimab		CTLA-4	Adenocarcinoma	Phase II	[40]
				NSCLC resistant anti-PD-1/PD-L1	Phase Ib/II	[41]
	Ivonescimab		VEGF	Advanced NSCLC	Phase I/II	[42,43]
Activate the immune system	Blinatumomab	CD3	CD19	R/R Ph −/+ B-ALL	Approved	[16,44,45]
				NHL		
	Glofitamab		CD20	R/R DLBCL	Approved	[46]
				R/R MCL	Phase I	[47]
	Teclistamab		BCMA	Follicular NHL and MM	Approved	[48]
	Catumaxomab		EpCAM	Intraperitoneal malignant ascites	Approved	[49,50]
	Tebentafusp		HLA-A*02:01	Uveal melanoma	Approved	[51,52]
	Acimtamig	CD16a	CD30	R/R PTCL	Phase II	[53,54,55]
	E5C1		HER2	Ovarian metastatic cancer	Preclinical	[56,57]

Abbreviations: DLBCL, diffuse large B-cell lymphoma; EGFR, epidermal growth factor receptor; HLA-A*02:01, A serotype is determined by the antibody’s recognition of the α2 domain of the HLA-A α-chain. For A*02, the α chain is encoded by the HLA-A*02 gene and the β chain is encoded by the B2M locus; MCL, mantle cell lymphoma; MM, multiple myeloma; NSCLN, non-small-cell lung cancer; NHL, non-Hodgkin lymphoma; R/R, relapsed or refractory; Ph −/+ B-ALL, Philadelphia chromosome-negative or -positive B cell acute lymphoblastic leukemia; PTCL, peripheral T-cell lymphoma.

**Table 2 biomolecules-15-00639-t002:** Main advantages and disadvantages of BsAbs.

Advantages	Disadvantages
High specificity	Some immunogenic
One molecule, two targets	Off-target effects
Block surface receptors	High production costs
Recruit cells	Poor tumor infiltration and penetration
Better than mAbs	Complex and huge
Activate and potentiate the immune system	Difficult cross-biological barriers

**Table 3 biomolecules-15-00639-t003:** Some examples of Nbs based on their application.

Nbs Use	Name	Target	Application	Phase	Reference
For imaging and diagnosis	^131^I-GMIB	HER2	PET	Phase I	[76]
	^18^F-AlF-RESCA-MIRC213			Preclinical	[77]
	^68^Ga-NODAGA-NM-01	PD-L1		Phase I	[78]
	^99m^Tc-PHG102	CLDN18.2	SPECT	Preclinical	[79]
	^99m^Tc-MY6349	Trop2		Phase I	[80]
	_99m_Tc-K2	PD-L1		Preclinical	[81]
	E8-IR800CW	CDH17	NIR	Preclinical	[82]
For cancer treatment	KN035	PD-L1	Solid tumors	Phase I/II	[83]
	Nb16	CTLA-4	Melanoma	Preclinical	[84]
	12A4	CXCL12	Pre-B lymphoma	Preclinical	[85]
	OA-cb6	EGFR	Epithelial cancers	Preclinical	[86]
	3VGR19	VEGFR	Angiogenesis in solid tumors	Preclinical	[87]

Abbreviations: NIR, near-infrared fluorescence imaging; PET, positron emission tomography; SPECT, single photon emission computed tomography.

**Table 4 biomolecules-15-00639-t004:** Main advantages and disadvantages of Nbs.

Advantages	Disadvantages
Low size	Short half-life time
High affinity, stability, solubility	Low biodisponibility
Low immunogenicity	Kidney accumulation
Low production costs	Some off-target effects
Well tumor infiltration and penetration	Immunogenicity
Unique physico-chemical properties	Treatment resistance
Easily modified	Difficulties crossing biological barriers

**Table 5 biomolecules-15-00639-t005:** Examples of EV-derived candidates in preclinical or clinical trials.

EVs’ Source	EV Engineering	Application	Phase	Reference
Fresh milk-derived exosomes	Yes, loaded with cisplatin	Resistant ovarian carcinoma	Preclinical	[114]
Ascite-derived exosomes	No	Colorectal cancer	Phase I	[115]
DC-derived exosomes	Yes, tumor antigen-loaded	NSCLC	Phase II	[116]
MVs from tumor cells	Yes, MTX packaging	Cholangiocarcinoma	Phase I	[118]
Plant exosomes	Yes, loaded with curcumin	Colon cancer	Phase I	[NCT01294072]
MSC-derived exosomes	Yes, KrasG12D siRNA added	Metastatic pancreatic cancer	Phase I	[117]
Grape-derived exosomes	No	Head and neck cancer	Phase I	[NCT01668849]

Abbreviations: DCs, dendritic cells; MSCs, mesenchymal stromal cells; MTX, methotrexate; NSCLN, non-small-cell lung cancer.

**Table 6 biomolecules-15-00639-t006:** Main advantages and disadvantages of EVs.

Advantages	Disadvantages
Biological origin, high biocompatible	Heterogeneity between EVs
Easily modified and functionalized	Source of EVs is relevant
Able to cross biological barriers	Production scaling-up
Different source available	Standardization methods for production
Natural delivery system	Loading is complex
High stability	Modifications may affect integrity
Low immunogenicity	

**Table 7 biomolecules-15-00639-t007:** Comparative of main properties of BsAbs, Nbs and EVs.

	BsAbs	Nbs	EVs
Size	150 kDa or 14 × 8 nm	15 kDa or 4 × 2.5 nm	50–500 nm
Origin	Usually, animals	Camelids and fishes	Plants and animal
Production complexity	Medium	Medium	High
Immunogenicity	High	Low	The lowest
Half-life	Medium	Low	High
Tumor penetration	Low	High	Medium
Production costs	High	Low	Medium
Stability	Medium	High	High
Off-target effects	Yes	Yes	Yes, but low
Cross biological barriers	Poor	Not easily	Easily
Activity	Intrinsic	Intrinsic	Not always

## Data Availability

Not applicable.

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
