# Peer review of "Bispecific Antibodies, Nanobodies and Extracellular Vesicles: Present and Future to Cancer Target Therapy"

_biomolecules, 2025, doi:10.3390/biom15050639_

Round 1
Reviewer 1 Report
Comments and Suggestions for Authors
- Is there any connection between mAb, BsAbs, and EVs? Specifically, the introduction should clarify whether there are mechanistic or functional connections among these modalities—such as potential synergistic effects, similar targeting mechanisms, or shared applications in cancer immunotherapy. Why did the authors only discuss EVs instead of other nanoparticles in this review? The authors should further highlight these in the introduction.
- Please discuss the capacity of tumor-secreted EVs for homotypic targeting and their potential use in cancer therapy. The source of EVs—whether derived from tumor cells, immune cells, or normal cell lines—has a significant impact on their targeting efficiency, immune compatibility, and therapeutic functions. The authors are encouraged to expand on how EV origin influences their applicability and effectiveness in cancer therapy.
- Tables summarizing representative preclinical and clinical studies involving mAbs, BsAbs, and EVs in cancer therapy are necessary
- Since the review focuses on both exosomes and microvesicles, it is recommended that the authors define and distinguish these two EV subtypes in the introduction.
- Can the authors give a comparative table for evaluating difference between BsAbs, Nbs, and EVs in cancer therapy
- The review should further discuss the clinical considerations and translational barriers associated with mAbs, BsAbs, and EVs. Critical issues include large-scale biomanufacturing, purification, formulation stability, storage conditions, and regulatory compliance (the authors have mentioned this part, which is commendable). The following references, all focused on clinical aspects, are recommended to strengthen this discussion: mAbs— org/10.1038/s41568-024-00690-x; BsAbs— doi.org/10.3389/fimmu.2021.626616; EVs—doi.org/10.1016/j.tibtech.2024.08.007.
Author Response
April 28th, 2025
Flossie Li
Assigned Editor,
Biomolecules
Dear Editor,
Thank you so much for allowing us to submit a revised version of our manuscript entitled “Bispecific antibodies, nanobodies and extracellular vesicles: present and future to cancer target therapy.” The authors thank the reviewers for their thoughtful comments and suggestions on our manuscript. We have considered all the comments and incorporated them into the revised version. Changes to the original document are highlighted as track changes, and an itemized point-by-point response to the reviewers’ comments is presented below.
COMMENTS FROM REVIEWER #1
Comment 1
Is there any connection between mAb, BsAbs, and EVs? Specifically, the introduction should clarify whether there are mechanistic or functional connections among these modalities—such as potential synergistic effects, similar targeting mechanisms, or shared applications in cancer immunotherapy. Why did the authors only discuss EVs instead of other nanoparticles in this review? The authors should further highlight these in the introduction.
Response 1: Thank you for pointing this out. Yes, the connection between them is that they can naturally occur and have a potential synergistic effect, as the reviewer mentioned in his/her comment. The change included a new paragraph in the manuscript, located on page 2 between line 83 and line 89, where there is given a more detailed explanation based on this.
We have decided to focus on EVs because there is a connection between antibodies and EVs, they have gained more attention to control or modify the tumor microenvironment (TME) in recent years and due to their biological origin and high biocompatibility to be use as a nanocarrier for drugs.
The paragraph now state “Recently, studies have demonstrated that B cells can secrete EVs that contain some kinds of antibodies on their surface and deliver their inside contents to the target cells [27-29]. Also, some authors have described that it is possible to encoated EVs with antibodies. The EVs were modified to achieve the expression of two types of monoclonal antibodies for dual targeting, so instead of two mAbs it is possible to use a BsAb or Nbs [30]. This potential combination of EVs with antibodies on their surface can better lead EVs to tumor cells and improve selectivity by reducing off-target effects.”
Comment 2
Please discuss the capacity of tumor-secreted EVs for homotypic targeting and their potential use in cancer therapy. The source of EVs—whether derived from tumor cells, immune cells, or normal cell lines—has a significant impact on their targeting efficiency, immune compatibility, and therapeutic functions. The authors are encouraged to expand on how EV origin influences their applicability and effectiveness in cancer therapy.
Response 2: Thanks for your comment. We have added a specific paragraph about that topic talking about the benefits that offered the different EVs sources and specially highlighting the capacity of tumor-derived EVs for homotypic targeting improving the delivery efficacy of anti-cancer drugs. The paragraph is situated on page 11, between lines 454 and 482 and now state “The source of EVs is also an important factor to consider. The most used cell sources include cancer cells, immune system cells, mesenchymal stem cells (MSCs), and some commercial cell lines. The right selection of the appropriate cell type is so important because EVs inherit most characteristics of parent cells, such as membrane characteristics and surface molecules, which influence their functions and effects on target cells [101]. EVs form immune cells retaining most of their parent cells’ characteristics such as immune, immunostimulatory and immunoregulatory properties that allow them to inhibit tumor proliferation, improve immune system activity and chemotherapeutic effects, induce cells to be high immunogenic by increasing recognition of ligands, and stimulate T cell antigen-specific responses. They are also able to be used as a delivery system for exogenous cargo to the target site [101-104]. Plant EVs are rich in bioactive components and have high biocompatibility, stability and biodegradability with low immunogenicity and toxicity and natural anti-inflammatory and immunomodulatory properties. For that reason, they are very useful for improving immunity function, modulating gene expression and for inducing tumor cell death by apoptosis as well as a delivery system for both hydrophilic and hydrophobic therapeutic agents [101,102,105,106]. EVs derived from animals are usually formed from milk and stand out for their abundance, easy extraction, safety and simply functionalization. They can be used for enhancing immune system activity as a delivery system and also for inhibiting tumor cell progression because they have inherent anti-cancer potential [101,102,107,108]. Tumor cell EVs (TEVs) have been focalized in the current research because they can be used for homotypic targeting, due to their same origin as tumor cells. This provides them with the ability to be self-recognized and absorbed easily, facilitating drug delivery to cancer cells and opening opportunities for personalized medicine [102,109]. This strategy improved significantly targeting tumor cells but not only this, also infiltration and cell apoptosis induction, reducing toxicity and slide effects on cancer treatment and reaching places that are hard to access such as the central nervous system. In recent years, there has been huge evidence that using them as drug carriers can improve the efficacy of cancer treatment [101,102,110].”
Comment 3
Tables summarizing representative preclinical and clinical studies involving mAbs, BsAbs, and EVs in cancer therapy are necessary
Response 3: Thank you for pointing this out. We agree with this comment and for that reason, we included some tables that summarize the most representative preclinical and clinical trials involving the three cancer therapeutic platforms. They are located on page 4, line 136 for BsAbs; on page 7, line 277 for Nbs; and, on page 12 line 507 for EVs. The tables included are the following ones.
Table 1. Summary of BsAbs that are approved or better positioned in clinical trials.
BsAbs mechanism of action |
Name |
Target 1 |
Target 2 |
Application |
Phase |
Reference |
Blocking two pathways |
Amivantamab |
EGFR |
MET |
NSCLC with EGFR mutations |
Approved |
[33] |
Vanucizumab |
VEGF |
Ang-2 |
Colorectal and metastatic cancers |
Phase I/II |
[34] |
|
Navicixizumab |
|
DLL4 |
Ovarian cancer |
Phase Ib |
[35] |
|
Zanidatamab |
HER2 |
HER2 |
Solid tumors |
Phase II |
[36] |
|
Against immune checkpoints |
Tebotelimab |
PD-1 |
LAG-3 |
Multiple cancer types |
Phase II/III |
[37,38] |
IBI318 |
|
PD-L1 |
NK/T-cell lymphoma |
Phase I/II |
[39] |
|
Cadonilimab |
|
CTLA-4 |
Adenocarcinoma |
Phase II |
[40] |
|
|
NSCLC resistant anti‐PD‐1/PD‐L1 |
Phase Ib/II |
[41] |
|||
Ivonescimab |
VEGF |
Advanced NSCLC |
Phase I/II |
[42,43] |
||
Activate the immune system |
Blinatumomab |
CD3 |
CD19 |
R/R Ph -/+ B-ALL |
Approved |
[16,44,45] |
NHL |
||||||
Glofitamab |
|
CD20 |
R/R DLBCL |
Approved |
[46] |
|
|
R/R MCL |
Phase I |
[47] |
|||
Teclistamab |
|
BCMA |
Follicular NHL and MM |
Approved |
[48] |
|
Catumaxomab |
|
EpCAM |
Intraperitoneal malignant ascites |
Approved |
[49,50] |
|
Tebentafusp |
|
HLA-A*02:01 |
Uveal melanoma |
Approved |
[51,52] |
|
Acimtamig |
CD16a |
CD30 |
R/R PTCL |
Phase II |
[53-55] |
|
E5C1 |
|
HER2 |
Ovarian metastatic cancer |
Preclinical |
[56,57] |
Abbreviations: DLBCL, diffuse large B-cell lymphoma; EGFR, epidermal growth factor receptor; MCL, mantle cell lymphoma; MM, multiple myeloma; NSCLN, non-small-cell lung cancer; NHL, non-Hodgkin lymphoma; R/R, relapsed or refractory; Ph -/+ B-ALL, Philadelphia chromosome negative or positive B cell acute lymphoblastic leukemia; PTCL, peripheral T-cell lymphoma.
Table 3. Some examples of Nbs based on their application.
Nbs use |
Name |
Target |
Application |
Phase |
Reference |
For imaging and diagnosis |
131I-GMIB |
HER2 |
PET |
Phase I |
[76] |
18F-AlF-RESCA-MIRC213 |
Preclinical |
[77] |
|||
68Ga-NODAGA-NM-01 |
PD-L1 |
Phase I |
[78] |
||
99mTc-PHG102 |
CLDN18.2 |
SPECT |
Preclinical |
[79] |
|
99mTc-MY6349 |
Trop2 |
Phase I |
[80] |
||
99mTc-K2 |
PD-L1 |
Preclinical |
[81] |
||
E8-IR800CW |
CDH17 |
NIR |
Preclinical |
[82] |
|
For cancer treatment |
KN035 |
PD-L1 |
Solid tumors |
Phase I/II |
[83] |
Nb16 |
CTLA-4 |
Melanoma |
Preclinical |
[84] |
|
12A4 |
CXCL12 |
Pre-B lymphoma |
Preclinical |
[85] |
|
OA-cb6 |
EGFR |
Epithelial cancers |
Preclinical |
[86] |
|
3VGR19 |
VEGFR |
Angiogenesis in solid tumors |
Preclinical |
[87] |
Abbreviations: NIR, near-infrared fluorescence imaging; PET, positron emission tomography; SPECT, single photon emission computed tomography.
Table 5. Examples of EVs-derived candidates in preclinical or clinical trials.
EVs’ source |
EV engineering |
Application |
Phase |
Reference |
Fresh milk derived exosomes |
Yes, loaded with cisplatin |
Resistant ovarian carcinoma |
Preclinical |
[114] |
Ascites-derived exosomes |
No |
Colorectal cancer |
Phase I |
[115] |
DCs-derived exosomes |
Yes, tumor antigen‐loaded |
NSCLC |
Phase II |
[116] |
MVs from tumor cells |
Yes, MTX packaging |
Cholangiocarcinoma |
Phase I |
[118] |
Plant exosomes |
Yes, loaded with curcumin |
Colon Cancer |
Phase I |
[NCT01294072] |
MSCs-derived exosomes |
Yes, KrasG12D siRNA added |
Metastatic pancreatic cancer |
Phase I |
[117] |
Grape-derived exosomes |
No |
Head and neck cancer |
Phase I |
[NCT01668849] |
Abbreviations: DCs, dendritic cells; MSCs, mesenchymal stromal cells; MTX, methotrexate; NSCLN, non-small-cell lung cancer.
Comment 4
Since the review focuses on both exosomes and microvesicles, it is recommended that the authors define and distinguish these two EV subtypes in the introduction.
Response 4: Thank you for your comment. We added a more detailed introduction about EVs, highlighting our decision to focus on them and we have also included a definition and a brief introduction about exosomes and microvesicles. These new paragraphs are situated on page 2, lines 75-82 and now state “According to their biogenesis and size we can distinguish four types: exosomes (50–200 nm); prostasomes (40–500 nm); microvesicles (MVs, 100–1,000 nm); and apoptotic bodies (50–4,000 nm) [26]. The two main types of EVs investigated and characterized are exosomes and MVs.
Exosomes originate from late endosomes and require endocytosis for its formation and exocytosis for its secretion; MVs do not require exocytosis because they are directly formed from the plasma membrane. They are produced and secreted by cancer cells, and they have a crucial role in tumor evasion and progression [5,20,22]. ”
Comment 5
Can the authors give a comparative table for evaluating difference between BsAbs, Nbs, and EVs in cancer therapy
Response 5: Thank you for pointing this out. We also agree with this comment. We have included a comparative table that recapitulates the differences between the three platforms. It is placed in section “5. Comparative Discussion”, “5.1. Comparing BsAbs, Nbs, and EVs in Cancer Therapy”, on page 14, line 564. The table included is the following one.
Table 7. Comparative of main properties of BsAbs, Nbs and EVs.
|
BsAbs |
Nbs |
EVs |
Size |
150 kDa or 14x8 nm |
15 kDa or 4x2.5 nm |
50-500 nm |
Origin |
Usually, animals |
Camelids and fishes |
Plants and animal |
Production complexity |
Medium |
Medim |
High |
Immunogenicity |
High |
Low |
The lowest |
Half-life |
Medium |
Low |
High |
Tumor penetration |
Low |
High |
Medium |
Production costs |
High |
Low |
Medium |
Stability |
Medium |
High |
High |
Off-target effects |
Yes |
Yes |
Yes, but low |
Cross biological barriers |
Poor |
Not easily |
Easily |
Activity |
Intrinsic |
Intrinsic |
Not always |
Comment 6
The review should further discuss the clinical considerations and translational barriers associated with mAbs, BsAbs, and EVs. Critical issues include large-scale biomanufacturing, purification, formulation stability, storage conditions, and regulatory compliance (the authors have mentioned this part, which is commendable). The following references, all focused on clinical aspects, are recommended to strengthen this discussion: mAbs— org/10.1038/s41568-024-00690-x; BsAbs— doi.org/10.3389/fimmu.2021.626616; EVs—doi.org/10.1016/j.tibtech.2024.08.007.
Response 6: Thanks to the reviewer for his/her comment. A new section was made according to clinical considerations and translational barriers (pages 15-16, lines 622-663), and now state “5.4. Clinical Considerations and Translational Barriers
BsAbs are frequently used in clinical diagnostics and treatment. For diagnosis, BsAbs can be combined with horseradish peroxidase, used in pre-targeting strategies to aid in clinical diagnosis, and provide better imaging to aid in the early detection, diagnosis, and treatment of cancer. Therapeutic strategies primarily aim to precisely target and reactivate immune cells, regulate their activation, fine-tune their fate and function, improve their tolerance, and promote a return to immune homeostasis. Besides treating tumors, BsAbs are also effective in treating hemophilia A, diabetes, Alzheimer's disease, and ophthalmological conditions [15]. There are several challenges associated with translational research of BsAbs: complex design and production, high immunogenicity risk, difficult analytical characterization, and an evolving regulatory environment.
Nbs are characterized by good tumor permeability and rapid renal clearance, which enables rapid and sensitive imaging of target tissue within hours of injection. As a result, they meet the criteria for an effective in vivo tracer [135-137]. In addition to their enhanced sensitivity, Nbs conjugates can deliver radionuclides to tumors, therefore allowing high-resolution as well as quantitative imaging of tumors. A further advantage of Nbs for nuclear imaging applications is their short biological half-life, comparable to PET isotopes [138,139]. Due to the multitude of advantages that Nbs possess over conventional mAbs and conventional mAb fragments, many alternate methods and techniques are being developed for producing application/situation-dependent Nbs. As a result, Nbs have found great success in various diagnostic formats, such as lateral flow immunoassay, diagnostic ELISAs, biosensors, and in vivo diagnostic imaging, not only for the detection of disease but also for the detection of food-borne pathogens and environmental toxins. Additionally, the multiple therapeutic approvals for Nbs treatment of cancer and autoimmune diseases have sparked commercial and industrial interest in Nbs over the past few years [17]. In order to conduct translational research on Nbs, several challenges must be overcome. Small size that is excellent for tissue penetration, rapid clearance: might require an extension of half-life and lower immunogenicity (camelid origin).
The clinical translation of EVs is still in progress, so there are several areas in which improvements could be made to ensure successful and timely translation into the clinic. Due to the limitations of current techniques for tracking and imaging EVs within cells and tissues, it would be beneficial to develop EV tracers that would improve our understanding of extracellular and intracellular trafficking, which is crucial for gaining mechanistic insights [140,141]. EV subpopulations may be characterized more accurately and rapidly using recently developed single particle analysis techniques, such as Exoview [142] and total internal reflection fluorescence microscopy [143]. Since the heterogeneity of EVs is an essential challenge for regulatory approval, nanoscale methods like nanoflow cytometry [144], imaging flow cytometry [145], and affinity-based methods [146] can be used to characterize and separate subpopulations of EVs to improve therapeutic and diagnostic capabilities and reduce heterogeneity. Translational research on EVs faces several challenges: Insufficient standardized isolation methods, difficult scaling-up, ambiguous potency of assays, and regulatory classification challenges.”
Reviewer 2 Report
Comments and Suggestions for Authors
This manuscript describes 3 auspicious cancer therapeutic platforms—bispecific antibodies, nanobodies, and extracellular vesicles, highlighting their current applications and future potential in a relevant manner, but it still lacks some quantitative comparisons. I recommend the manuscript for publication after some minor revisions.
- Keywords should be ordered alphabetically.
- The article would benefit from a table with the pros and cons of each type of therapeutic agent.
- The article would also benefit from a table comparing size, production complexity, immunogenicity, half-life, tumor penetration, etc.
- Linex 197-198: I think there are some repeated words.
- Regarding the current applications in cancer therapy of all three, I would like to see some imaging/other images of real applications in cancer therapy, maybe from other original/research articles, showing the real therapeutic effect and some description of the results found in these articles.
- Line 482: I would like to see some images with the results obtained by the combinations described.
- The description of Figure 2 needs to be more detailed, and it also needs to be mentioned earlier in the manuscript
- There should be a discussion of how these therapies might be used in personalized medicine, maybe in the future perspectives section.
Author Response
April 28th, 2025
Flossie Li
Assigned Editor,
Biomolecules
Dear Editor,
Thank you so much for allowing us to submit a revised version of our manuscript entitled “Bispecific antibodies, nanobodies and extracellular vesicles: present and future to cancer target therapy.” The authors thank the reviewers for their thoughtful comments and suggestions on our manuscript. We have considered all the comments and incorporated them into the revised version. Changes to the original document are highlighted as track changes, and an itemized point-by-point response to the reviewers’ comments is presented below.
COMMENTS FROM REVIEWER #2
General comment
This manuscript describes 3 auspicious cancer therapeutic platforms—bispecific antibodies, nanobodies, and extracellular vesicles, highlighting their current applications and future potential in a relevant manner, but it still lacks some quantitative comparisons. I recommend the manuscript for publication after some minor revisions.
Response: Thanks to the reviewer for his/her kind comment about our manuscript.
Comment 1
Keywords should be ordered alphabetically.
Response 1: Thanks to the reviewer for pointing this out. The keywords were alphabetically ordered. It now state “Keywords: bispecific antibodies; cancer therapy; extracellular vesicles; nanobodies; target therapy.”
Comment 2
The article would benefit from a table with the pros and cons of each type of therapeutic agent.
Response 2: Thank you for comment. We agree with this and for that reason, we included some tables that summarize the main advantages and disadvantages of each cancer therapeutic platform. They are located on page 6, line 230 for BsAbs; on page 9, line 384 for Nbs; and, on page 13 line 532 for EVs. The tables included are the following ones.
Table 2. Main advantages and disadvantages of BsAbs.
Advantages |
Disadvantages |
High specificity |
Some immunogenic |
One molecule, two targets |
Off-target effects |
Block surface receptors |
High production costs |
Recruit cells |
Poor tumor infiltration and penetration |
Better than mAbs |
Complex and huge |
Activate and potentiate the immune system |
Difficult cross biological barriers |
Table 4. Main advantages and disadvantages of Nbs.
Advantages |
Disadvantages |
Low size |
Short half-life time |
High affinity, stability, solubility |
Low biodisponibility |
Low immunogenicity |
Kidney accumulation |
Low production costs |
Some off-target effects |
Well tumor infiltration and penetration |
Immunogenicity |
Unique physico-chemical properties |
Treatment resistance |
Easily modified |
Difficulties crossing biological barriers |
Table 6. Main advantages and disadvantages of EVs.
Advantages |
Disadvantages |
Biological origin, high biocompatible |
Heterogeneity between EVs |
Easily modified and functionalized |
Source of EVs is relevant |
Able to cross biological barriers |
Production scaling-up |
Different source available |
Standardization methods for production |
Natural delivery system |
Loading is complex |
High stability |
Modifications may affect integrity |
Low immunogenicity |
|
Comment 3
The article would also benefit from a table comparing size, production complexity, immunogenicity, half-life, tumor penetration, etc.
Response 3: Thanks for your comment. We also agree that the article is benefit with a table comparing some properties of the three platforms. We added a comparative table that recapitulates the differences between them, and it is placed in section “5. Comparative Discussion”, “5.1. Comparing BsAbs, Nbs, and EVs in Cancer Therapy”, on page 14, line 564. The table included is the following one.
Table 7. Comparative of main properties of BsAbs, Nbs and EVs.
|
BsAbs |
Nbs |
EVs |
Size |
150 kDa or 14x8 nm |
15 kDa or 4x2.5 nm |
50-500 nm |
Origin |
Usually, animals |
Camelids and fishes |
Plants and animal |
Production complexity |
Medium |
Medim |
High |
Immunogenicity |
High |
Low |
The lowest |
Half-life |
Medium |
Low |
High |
Tumor penetration |
Low |
High |
Medium |
Production costs |
High |
Low |
Medium |
Stability |
Medium |
High |
High |
Off-target effects |
Yes |
Yes |
Yes, but low |
Cross biological barriers |
Poor |
Not easily |
Easily |
Activity |
Intrinsic |
Intrinsic |
Not always |
Comment 4
Linex 197-198: I think there are some repeated words.
Response 4: Thanks to the reviewer for pointing this out. We agree with this comment. It is true that there were some repeated words that were eliminated. The sentence now state “Nanobodies (Nbs) are also called single-domain antibodies (sdAbs) or variable domain of heavy-chain antibodies (VHH) because they are composed of a scFv derived from the heavy-chain antibodies of the Camelidae family (usually, camels and llamas) and some fish (in particular, sharks) [16,19,71,72].”
Comment 5
Regarding the current applications in cancer therapy of all three, I would like to see some imaging/other images of real applications in cancer therapy, maybe from other original/research articles, showing the real therapeutic effect and some description of the results found in these articles.
Response 5: Thanks to the reviewer for his/her comment. We have highlighted the following reference, Liu, X.; Xiao, C.; Xiao, K. Engineered extracellular vesicles-like biomimetic nanoparticles as an emerging platform for targeted cancer therapy. J Nanobiotechnology 2023, 21, 287, doi:10.1186/s12951-023-02064-1, and we have added A new section was made according to clinical considerations and translational barriers (pages 15-16, lines 622-663), and now state “5.4. Clinical Considerations and Translational Barriers
BsAbs are frequently used in clinical diagnostics and treatment. For diagnosis, BsAbs can be combined with horseradish peroxidase, used in pre-targeting strategies to aid in clinical diagnosis, and provide better imaging to aid in the early detection, diagnosis, and treatment of cancer. Therapeutic strategies primarily aim to precisely target and reactivate immune cells, regulate their activation, fine-tune their fate and function, improve their tolerance, and promote a return to immune homeostasis. Besides treating tumors, BsAbs are also effective in treating hemophilia A, diabetes, Alzheimer's disease, and ophthalmological conditions [15]. There are several challenges associated with translational research of BsAbs: complex design and production, high immunogenicity risk, difficult analytical characterization, and an evolving regulatory environment.
Nbs are characterized by good tumor permeability and rapid renal clearance, which enables rapid and sensitive imaging of target tissue within hours of injection. As a result, they meet the criteria for an effective in vivo tracer [135-137]. In addition to their enhanced sensitivity, Nbs conjugates can deliver radionuclides to tumors, therefore allowing high-resolution as well as quantitative imaging of tumors. A further advantage of Nbs for nuclear imaging applications is their short biological half-life, comparable to PET isotopes [138,139]. Due to the multitude of advantages that Nbs possess over conventional mAbs and conventional mAb fragments, many alternate methods and techniques are being developed for producing application/situation-dependent Nbs. As a result, Nbs have found great success in various diagnostic formats, such as lateral flow immunoassay, diagnostic ELISAs, biosensors, and in vivo diagnostic imaging, not only for the detection of disease but also for the detection of food-borne pathogens and environmental toxins. Additionally, the multiple therapeutic approvals for Nbs treatment of cancer and autoimmune diseases have sparked commercial and industrial interest in Nbs over the past few years [17]. In order to conduct translational research on Nbs, several challenges must be overcome. Small size that is excellent for tissue penetration, rapid clearance: might require an extension of half-life and lower immunogenicity (camelid origin).
The clinical translation of EVs is still in progress, so there are several areas in which improvements could be made to ensure successful and timely translation into the clinic. Due to the limitations of current techniques for tracking and imaging EVs within cells and tissues, it would be beneficial to develop EV tracers that would improve our understanding of extracellular and intracellular trafficking, which is crucial for gaining mechanistic insights [140,141]. EV subpopulations may be characterized more accurately and rapidly using recently developed single particle analysis techniques, such as Exoview [142] and total internal reflection fluorescence microscopy [143]. Since the heterogeneity of EVs is an essential challenge for regulatory approval, nanoscale methods like nanoflow cytometry [144], imaging flow cytometry [145], and affinity-based methods [146] can be used to characterize and separate subpopulations of EVs to improve therapeutic and diagnostic capabilities and reduce heterogeneity. Translational research on EVs faces several challenges: Insufficient standardized isolation methods, difficult scaling-up, ambiguous potency of assays, and regulatory classification challenges.”
Comment 6
Line 482: I would like to see some images with the results obtained by the combinations described.
Response 6: Thanks to the reviewer for his/her comment. We have highlighted the following reference, Liu, X.; Xiao, C.; Xiao, K. Engineered extracellular vesicles-like biomimetic nanoparticles as an emerging platform for targeted cancer therapy. J Nanobiotechnology 2023, 21, 287, doi:10.1186/s12951-023-02064-1. With emphasis in the Figure 7. Please find enclosed the figure.
Figure 7. Liu, X.; Xiao, C.; Xiao, K. Engineered extracellular vesicles-like biomimetic nanoparticles as an emerging platform for targeted cancer therapy. J Nanobiotechnology 2023, 21, 287, doi:10.1186/s12951-023-02064-1.
Comment 7
The description of Figure 2 needs to be more detailed, and it also needs to be mentioned earlier in the manuscript.
Response 7: Thanks to the reviewer for his/her comment, the Figure was modified, making it more detailed and moved (page 13, lines 534-538).
Figure 1. The present and future use of BsAbs, Nbs, and EVs in cancer therapy.
Comment 8
There should be a discussion of how these therapies might be used in personalized medicine, maybe in the future perspectives section.
Response 8: Thanks to the reviewer for his/her comment. Novel information was added in the final section (pages 16-17, lines 674-691), with special emphasis on the application of these three platforms in personalized medicine and now state “A transformative step in the future of cancer therapy will be the integration of BsAbs, Nbs, and EVs into personalized medicine. We are beginning to understand that one-size-fits-all approaches are insufficient for optimal clinical outcomes as we gain a deeper understanding of tumor heterogeneity. The ability of BsAbs to engage both tumor-specific antigens and immune effector cells offers a promising platform for designing highly tailored treatments based on the molecular profile of a patient's tumor. The advantages of Nbs include enhanced tissue penetration, reduced immunogenicity, and ease of genetic engineering, making them ideal candidates for patient-specific targeting.
Furthermore, EVs are gaining attention as natural, biocompatible carriers for the delivery of antibody fragments, RNA molecules, or therapeutic payloads directly to the TME. Due to their inherent ability to reflect the biological state of their parent cells, they are potential tools not only for therapeutic use, but also for diagnostic and monitoring purposes. With the convergence of these technologies and high-throughput omics data, as well as AI-driven patient stratification tools, we may be able to develop customized therapeutic regimens which maximize efficacy while minimizing adverse effects.
These biologic therapies are poised to play an increasingly important role in the realization of truly personalized cancer care, providing patients with precision, adaptability, and improved outcomes.”
Round 2
Reviewer 1 Report
Comments and Suggestions for Authors
The authors have satisfactorily addressed my concerns!